# Jungle Express is a versatile repressor system for tight transcriptional control

Thomas L. Ruegg[1,2], Jose H. Pereira[1,3], Joseph C. Chen[1,4], Andy DeGiovanni[1,3], Pavel Novichkov[5], Vivek K. Mutalik[5], Giovani P. Tomaleri[1,3], Steven W. Singer [1,6], Nathan J. Hillson [1,6], Blake A. Simmons [1,6], Paul D. Adams[1,3,7] & Michael P. Thelen[1,8]

Tightly regulated promoters are essential for numerous biological applications, where strong inducibility, portability, and scalability are desirable. Current systems are often incompatible with large-scale fermentations due to high inducer costs and strict media requirements. Here, we describe the bottom-up engineering of 'Jungle Express', an expression system that enables efficient gene regulation in diverse proteobacteria. This system is guided by EilR, a multidrug-binding repressor with high affinity to its optimized operator and cationic dyes that act as powerful inducers at negligible costs. In *E. coli*, the engineered promoters exhibit minimal basal transcription and are inducible over four orders of magnitude by 1 μM crystal violet, reaching expression levels exceeding those of the strongest current bacterial systems. Further, we provide molecular insights into specific interactions of EilR with its operator and with two inducers. The versatility of Jungle Express opens the way for tightly controlled and efficient gene expression that is not restricted to host organism, substrate, or scale.

[1] Joint BioEnergy Institute, Emeryville, CA 94608, USA. [2] Institute of Botany, University of Basel, 4001 Basel, Switzerland. [3] Molecular Biophysics and Integrated Bioimaging Division, Lawrence Berkeley National Laboratory, Berkeley, CA 94720, USA. [4] Department of Biology, San Francisco State University, San Francisco, CA 94132, USA. [5] Environmental Genomics and Systems Biology Division, Lawrence Berkeley National Laboratory, Berkeley, CA 94720, USA. [6] Biological Systems and Engineering Division, Lawrence Berkeley National Laboratory, Berkeley, CA 94720, USA. [7] Department of Bioengineering, University of California Berkeley, Berkeley, CA 94720, USA. [8] Biology and Biotechnology Division, Lawrence Livermore National Laboratory, Livermore, CA 94550, USA. These authors contributed equally: Jose H. Pereira, Joseph C. Chen. Correspondence and requests for materials should be addressed to M.P.T. (email: thelen1@llnl.gov)

Bacteria have evolved diverse mechanisms to sense and adapt to changes in the environment. Typically, these responses are mediated at the transcriptional level by allosteric transcription factors, DNA-binding proteins that establish specific contacts with a chemical signal in the ligand-binding domain. The interaction triggers a conformational change that transduces the signal to the DNA-binding domain; this in turn changes its affinity to a specific operator within the promoter region, which results in differential gene expression. The decoupling of regulatory modules from their native context provides a means to gain external control over transcription of target genes by the addition of their inducing effector molecules.

Inducible promoters are indispensable tools in biological research, including the study of gene function and cellular regulation, and the engineering of strains with new capabilities. Tight transcriptional control is essential, since basal expression levels can obscure the interpretation of phenotypic studies and cause detrimental off-target reactions in bioengineering projects[1]. Inducible systems are also vital for biotechnological applications, including high-level gene expression for the production of enzymes and therapeutic proteins, and the regulation of metabolic pathways for the biosynthesis of pharmaceuticals, fragrances, nutraceuticals, chemical building blocks, and biofuels[2–4]. While strong promoters are often required in such applications, gene overexpression also causes a general metabolic burden on cells[5], and heterologous expression is often toxic to the host organism[3,6]. These stresses result in reduced growth rates and increase the risk of plasmid loss and escape mutants when using constitutive or leaky promoters[7], ultimately leading to poor productivities[2,6,8]. Such detrimental effects can be minimized by using a tightly repressible induction system that completely decouples growth from production, which allows for the establishment of a fast-growing, healthy culture prior to expression of target genes during the production phase[6].

There is a long history of resourcing regulatory parts from nature and refactoring them as inducible gene expression systems. Major developments that employed transcriptional machineries from phages are now routinely used for high-level gene expression[9,10]. The output of induction systems has been varied by combining operator sites with phage promoters and by selecting suitable operator/promoter hybrids out of randomized libraries[11,12]. The increasing knowledge of transcription factors in combination with progress in DNA sequencing and synthesis has promoted the mining of genomic databases and the screening for ligands to discover novel regulatory systems for applications[13,14]. Despite the sizeable knowledgebase of transcription factors, their cognate operator sites and corresponding ligands, only a few induction mechanisms, including the LacI-IPTG, the TetR-aTc, and AraC-arabinose systems, are routinely used in practice[2,10]. Depending on the application, these systems often exhibit limitations, such as high basal expression, narrow host range, or specific growth requirements. Media components might interfere with conventional inducible systems, resulting in loss of regulation and poor performance. For example, the presence galactose, arabinose, or rhamnose in plant biomass-based fermentations[15] reduces the regulatory effect of common promoters induced by these monosaccharides[16–18], while glucose concomitantly prevents full activation due to carbon catabolite repression[19,20]. Such considerations become critical when scaling up from culture tube to large bioreactors, where high efficiency and low operating costs are essential[3,4,6]. In particular, the high costs of current inducers usually preclude their use in industrial-sized fermentations. Consequently, suboptimal constitutive promoters are often standard practice, at the expense of high productivities that are particularly required for generating low-cost products[3,6,21].

Alternative induction strategies include the use of promoters that are activated by cell density-dependent signals[22], by starving cells of an essential nutrient[23], or by dynamic pathway regulation controlled by intermediates[24]. Although these systems do not depend on the addition of inducing compounds, the timing and level of expression are generally difficult to control, and in many instances reduce metabolic activity and require host engineering or stringent media compositions[25,26]. In contrast, an ideal inducible system would exhibit minimal basal activity, display strong and uniform expression levels upon induction, operate at low costs, and not be influenced by the host metabolism, media components, and other inducible systems.

Here, we describe the development and the applicability of a bacterial broad-host expression system that is inducible and displays minimal basal transcription. We previously identified EilR as a regulatory component of a multidrug efflux system from *Enterobacter lignolyticus*[27], a bacterium isolated from the soil of a Puerto Rican rainforest because of its ability to catabolize lignin components[28]. In its native context, EilR regulates expression of EilA, an inner membrane transporter that confers tolerance to imidazolium-based ionic liquids, reagents that enhance the microbial conversion of lignocellulosic biomass to chemicals. The ability of EilR to respond to these reagents provides a substrate-responsive, auto-regulated tolerance system that maintains its functionality in a biofuel-producing *E. coli* strain. In this work, we generated an operator with increased affinity for EilR by comparing conserved sequences across multiple bacterial genomes. We then combined this operator with *E. coli*-phage immediate-early promoters that are recognized by the host RNA polymerase[29]. By using an EilR-regulated promoter probe, we identified several cationic dyes that act as efficient low-cost inducers. These molecules bind to EilR with high affinities, capable of releasing the repressor from its operator at nM to µM concentrations in *E. coli* and three distantly related proteobacteria. Using data from X-ray crystallography, we present insights on EilR interaction with its operator and identify contacts with two inducers, crystal violet (CV) and malachite green (MG). Alluding to the source of EilR from a rainforest bacterium, we named the resulting induction system "Jungle Express" (JEx).

## Results

**A palindromic consensus operator increases affinity to EilR.** EilR belongs to the TetR family of transcription factors, which commonly regulate divergently transcribed adjacent genes[30]. This inferred that the intergenic region between *eilR* and the *eilA* efflux pump genes contains cognate EilR binding sites, the eil-operators (*eilO*) (Fig. 1a). By searching for motifs in regions located between *eilR* homologs and divergently aligned *eilA* homologs in gamma-proteobacteria, we identified a 24-bp consensus motif *eilO_c*, consisting of two conserved, inverted 11-bp sequences separated by two base pairs that is represented twice in these intergenic regions. In *E. lignolyticus*, this palindromic motif is located 56–79 bp (*eilO_1*) and 20–43 bp (*eilO_2*) upstream of *eilA*, with *eilO_1* embedded in both the *eilR* and *eilA* promoter regions (Fig. 1a).

Using an electrophoretic mobility shift assay (EMSA) to test the affinity of purified EilR to each of the two native DNA sequences and to the consensus operator, we confirmed their function as EilR binding sites. In particular, we observed that *eilO_c* binds with high affinity to EilR, exceeding that of the native *E. lignolyticus* operators (Fig. 1b).

**Cationic dyes release EilR from its operator.** To identify potential EilR ligands, we created a reporter plasmid containing

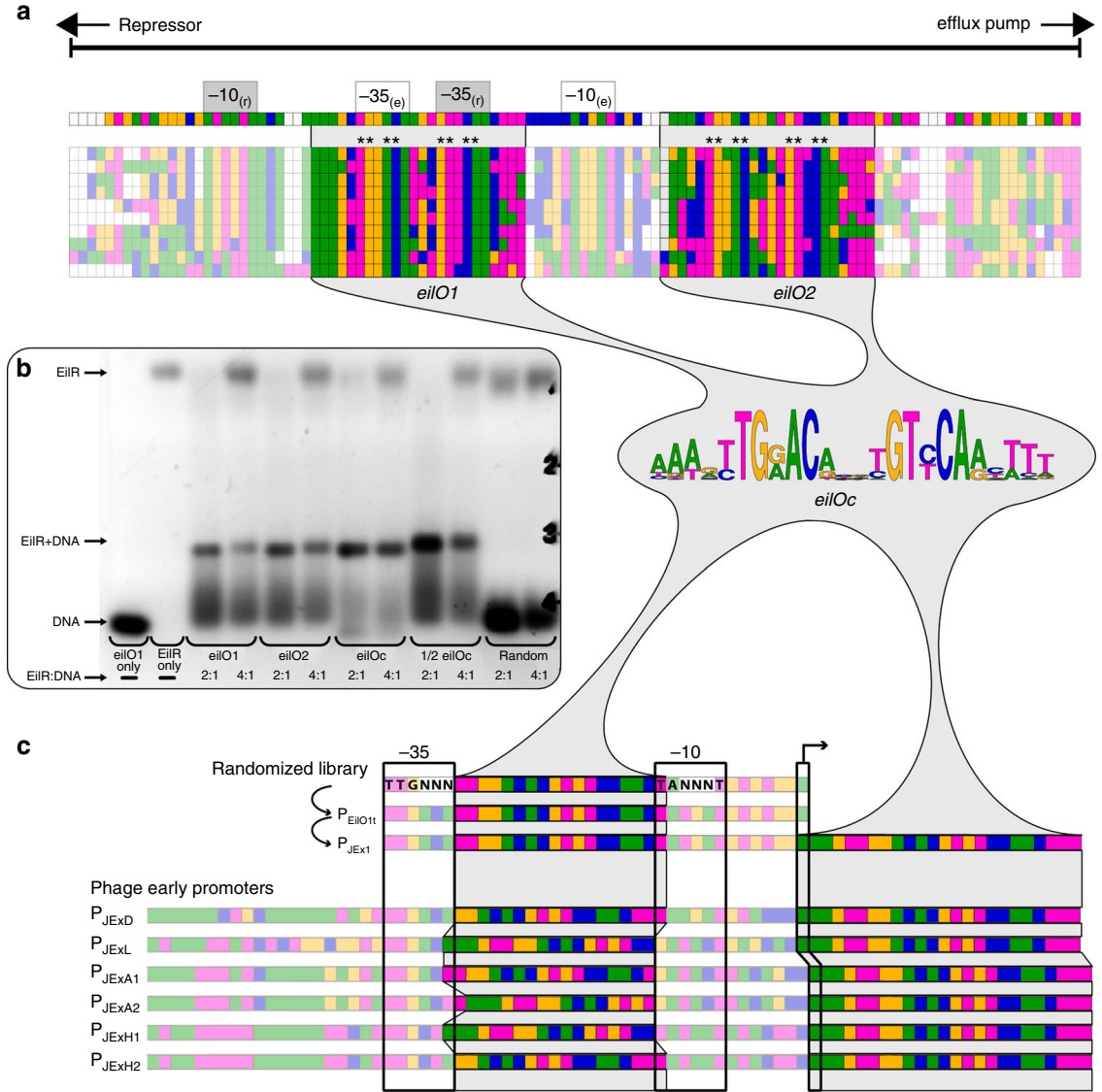

**Fig. 1** EilR repressor-regulated promoter engineering. **a** Alignment of intergenic regions that separate the genes encoding efflux pumps and their cognate repressors in gamma-proteobacteria homologous to the *E. lignolyticus eilAR* locus. All intergenic regions contain two repressor-binding sites *eilO₁* and *eilO₂*, from which the 24-bp consensus operator (*eilO_c*) motif emerges. The asterisks indicate the base pairs conserved in all operators. Predicted promoter hexamers are shown for the repressor genes ($-35_r$, $-10_r$) and for the efflux pumps ($-35_e$, $-10_e$). Each colored box represents a nucleotide: A (green); T (magenta); G (yellow); C (blue). **b** Electrophoretic mobility shift binding assays of purified EilR with the full-length native *E. lignolyticus* operators (*eilO₁*, *eilO₂*), the full consensus operator (*eilO_c*), half consensus operator (½ *eilO_c*) and random DNA. Molar ratios of the 21.6 kDa EilR monomer and duplex DNA are indicated. **c** A library of randomized *E. coli* consensus promoter boxes fused with a truncated consensus operator generated the biosensor P_EilO1t, into which an additional full-length *eilO_c* was placed at the transcriptional start site to yield P_JEx1. Immediate-early coliphage-promoters P_D/E20 (P_JExD) and P_H207 (P_JExH1, P_JExH2) from phage T5, P_L (P_JExL) from phage lambda, and P_A1 (P_JExA1, P_JExA2) from phage T7, were reorganized by placing a truncated *eilO_c* into the spacer region, partially overlapping the −35 or the −10 hexamers, followed by addition of a full-length consensus operator at the transcriptional start site. An arrow indicates the transcriptional start site. Colors of the nucleotides belonging to the *eilO_c* -operator are highlighted and the promoter −35 and −10 hexamers are boxed

a constitutively driven *eilR*, and a truncated *eilO_c* flanked by randomized promoter hexamers (−35 and −10 sites) upstream of the gene encoding the red fluorescent protein (RFP). *E. coli* carrying this randomized promoter library was then screened in the presence of the known EilR effector 2-ethyl-1-methylimidazolium chloride[27] to isolate promoter P_EilO1t, which showed the highest RFP expression level in response to this effector (Supplementary Fig. 1).

Next, the P_EilO1t-carrying reporter strain was exposed to three other known substrates of the cognate multidrug efflux pump EilA[27], all being hydrophobic ammonium cations.

The long-chained cetylpyridinium chloride and the bivalent cation methyl viologen caused only minimal de-repression at sublethal concentrations (Supplementary Fig. 2). In contrast, the acridine dye proflavine induced the reporter to a higher RFP expression level at a concentration ~10⁴ fold lower than that required for maximally achievable induction by 2-ethyl-1-methylimidazolium chloride. Given the sensitive response of EilR to μM levels of proflavine, we expanded the screen to other readily available hydrophobic cationic dyes, some of which are known to interact with the multidrug-binding repressors QacR[31] and RamR[32].

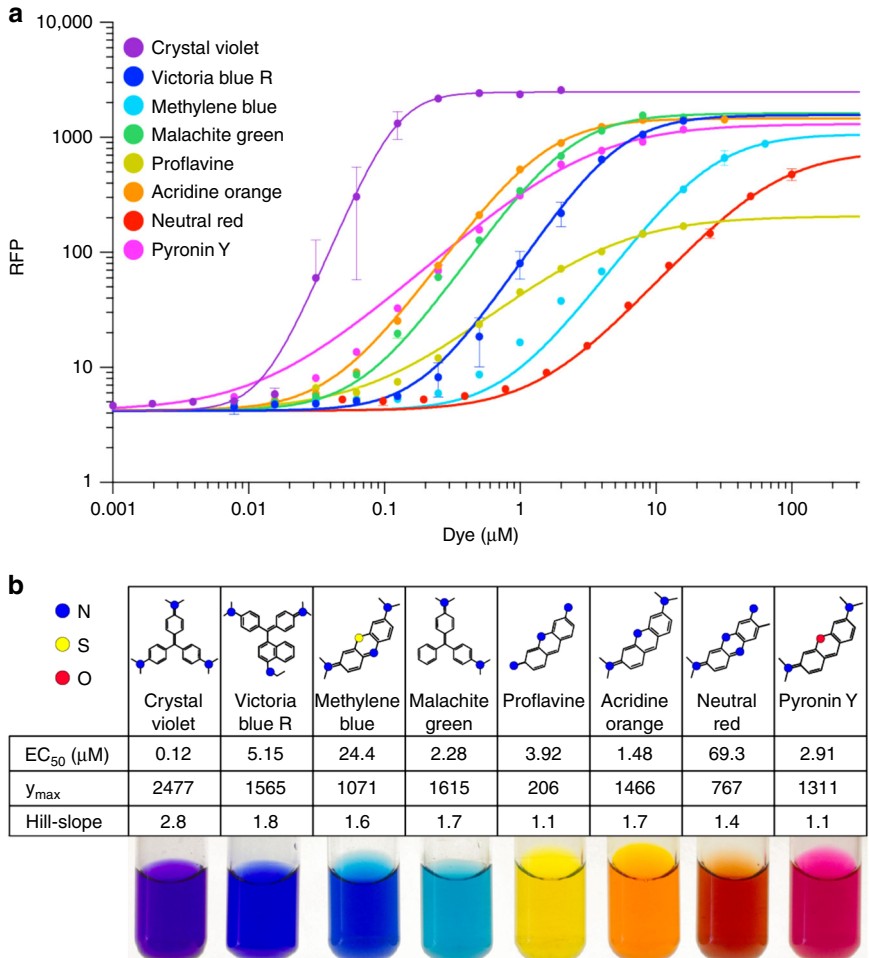

**Fig. 2** Cationic dyes induce EilR-regulated promoters. **a** Response of P$_{JExD}$ to the inducing dyes as measured by single-cell RFP fluorescence of *E. coli* containing the low-copy plasmid pTR_sJExD-rfp. **b** Values for EC$_{50}$, amplitudes (y$_{max}$), and Hill-coefficients are used to characterize the response of P$_{JExD}$ to cationic dyes. Colored circles in the chemical structures indicate N (blue), S (yellow), and O (red). Photographs show the corresponding inducing dyes in concentrated aqueous solutions. Values and error bars in **a** represent the means and standard deviation of biological triplicate measurements. Response curves (R$^2$ > 0.99) and numerical values in **b** were obtained by non-linear regression analysis in a 4-parameter variable slope model using average values of fluorescence measurements from three independently grown cultures

We found that several members of the acridine, phenothiazine, phenazine, and xanthene families induced the EilR-regulated reporter in the nM to low µM range (Fig. 2). Each of the identified effectors triggered a distinct promoter response. For example, increasing the concentration of acridine orange or of pyronin Y resulted in a relatively gradual induction. In contrast, crystal violet (CV) rapidly induced transcription, indicating a strong positive cooperativity[33]. Since CV also displayed the highest potency (EC$_{50}$ of 120 nM) and efficacy of all compounds tested (Fig. 2), we chose this purple triarylmethane dye as standard inducer in further experiments.

**Engineered promoters are inducible in diverse proteobacteria.** The high affinity of EilR for its consensus operator, as well as for CV motivated us to develop an EilR-regulated bacterial expression system. We chose a set of immediate-early promoters from *E. coli* phages[29], namely P$_{D/E20}$ and P$_{H207}$ from phage T5, P$_{A1}$ from phage T7, and P$_L$ from phage lambda, all of which are recognized by the host RNA-polymerase. Analogously to P$_{EilO1t}$, we first placed the truncated *eilO$_c$*-operator site into the 17-bp spacer region between and partly overlapping the −35 and −10 transcription motifs (Fig. 1c).

Insertion of a full-length second consensus operator at the transcriptional start site not only enhanced repression, but also elevated RFP levels in the induced state (Supplementary Fig. 3). The higher protein level is a likely consequence of increased transcript abundance, since the palindromic full-length operator has the potential to stabilize mRNA by forming a strong 5′-terminal stem-loop[34]. In the absence of inducer, basal activity of these EilR-regulated promoters fell below that of the three routinely used inducible systems P$_{BAD}$, P$_{tet}$, and P$_{trc}$ (Fig. 3). Both in complex and defined media with glucose as carbon source, most P$_{JEx}$-promoters exhibited approximately 10-fold stronger repression than that of the tightly regulated P$_{tet}$ and the arabinose-responsive, glucose-repressible P$_{BAD}$ promoter[10,35] (Fig. 3a, b). While basal RFP expression was barely detectable in the repressed state, CV induced P$_{JExD}$ in a population-uniform manner (Fig. 3c), resulting in more than a 10$^4$-fold dynamic range in both media tested.

We investigated whether the EilR-mediated promoter P$_{JEx1}$ interferes with these three inducible systems and their cognate inducer molecules arabinose, anhydrotetracycline, and IPTG, respectively (Supplementary Fig. 4). The absence of crosstalk demonstrates that EilR-based promoters are suitable for ortho-gonal gene regulation.

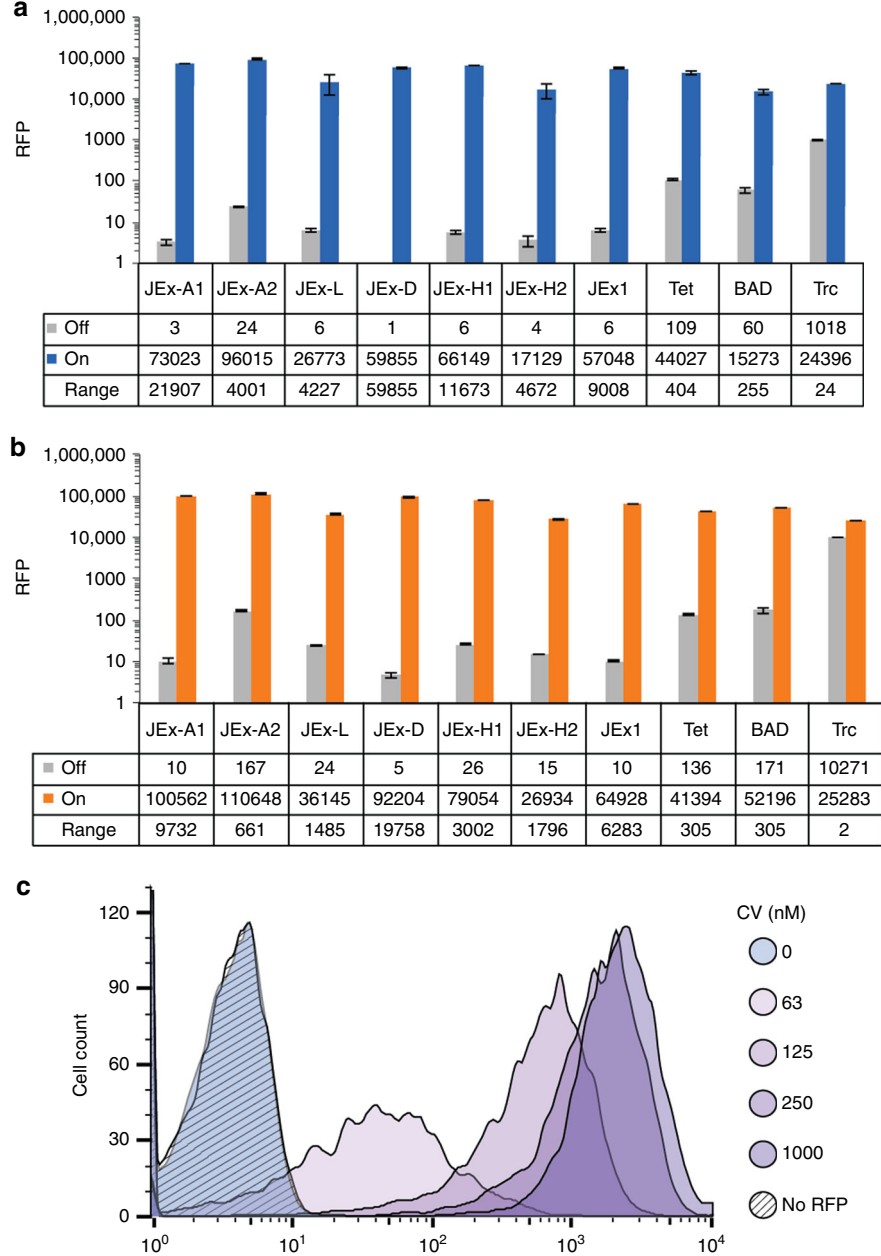

**Fig. 3** RFP expression from EilR-regulated promoters in *E. coli*. Single-cell fluorescence of *E. coli* expressing RFP from EilR-regulated and three common inducible promoters after growth in either defined media with 0.2% glucose (**a**) or LB media (**b**). $P_{JEx}$ promoters were induced with 1 μM CV, $P_{tet}$ with 400 nM anhydrotetracycline, $P_{BAD}$ with 13 mM arabinose and $P_{trc}$ with 1 mM IPTG. **c** Flow cytometry measurements show the distribution of single-cell fluorescence of stationary phase populations expressing RFP from $P_{JExD}$ after growth without (blue) or with increasing concentrations of CV (purple). The hatched histogram represents autofluorescence of cells lacking *rfp*. *E. coli* DH10B was used in **a–c**, with *rfp* expressed from medium copy plasmids (p15A *ori*). Values and error bars in **a** and **b** represent the means and standard deviation of biological triplicate measurements after subtracting background fluorescence emitted by *E. coli* lacking *rfp*

To test the host range of our system, we introduced the unmodified cassette comprising *eilR* and the suite of $P_{JEx}$ promoters into three non-enteric model proteobacteria: the metabolically versatile soil bacterium *Pseudomonas putida* KT2440[36]; the $N_2$-fixing plant symbiont *Sinorhizobium meliloti* Rm1021[37]; and the aquatic oligotroph *Caulobacter crescentus* NA1000[38]. In each of these hosts, EilR maintained its repressing capability and CV induced all of the examined promoters (Fig. 4). In these three bacteria, most EilR-regulated promoters displayed lower basal activities and higher expression maxima

than those of the TetR-dependent phage promoter $P_{LtetO-1}$[11]. In the gamma-proteobacteria *P. putida* and *E. coli*, $P_{JExD}$ exhibited the lowest basal activity and very high levels of expression in its induced state. Likewise, in the phylogenetically more distant *S. meliloti* and *C. crescentus*, this promoter exhibited relatively low basal expression and high activity when induced. However, in these two alpha-proteobacteria, $P_{JExH1}$ and $P_{JExH2}$ maintained tighter repression that resulted in larger dynamic ranges, even though they did not achieve expression maxima as high as that of $P_{JExD}$.

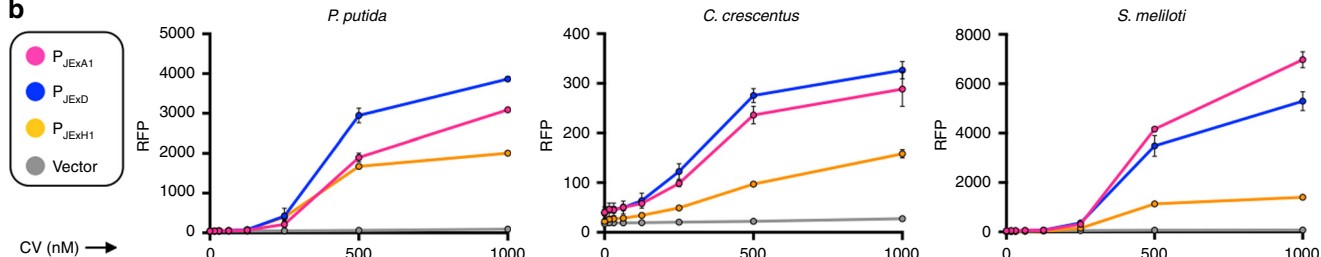

| | | P$_{JEx-A1}$ | P$_{JEx-A2}$ | P$_{JEx-L}$ | P$_{JEx-D}$ | P$_{JEx-H1}$ | P$_{JEx-H2}$ | P$_{JEx1}$ | P$_{LtetO}$ | Vector |
|---|---|---|---|---|---|---|---|---|---|---|
| *P. putida* | Uninduced | 42 (1) | 79 (13) | 43 (2) | 41 (1) | 45 (1) | 43 (1) | 56 (3) | 67 (2) | 42 (1) |
| | Induced | 3088 (34) | 3408 (98) | 2098 (123) | 3866 (4) | 2005 (13) | 522 (19) | 4342 (350) | 1218 (31) | 92 (5) |
| *C. crescentus* | Uninduced | 40 (7) | 36 (8) | 44 (5) | 38 (13) | 22 (3) | 22 (3) | 59 (9) | 50 (4) | 17 (3) |
| | Induced | 288 (34) | 253 (21) | 208 (19) | 326 (17) | 158 (8) | 177 (27) | 248 (37) | 233 (10) | 28 (2) |
| *S. meliloti* | Uninduced | 46 (2) | 84 (2) | 64 (5) | 35 (2) | 14 (1) | 19 (1) | 119 (3) | 41 (2) | 13 (1) |
| | Induced | 6977 (323) | 7089 (249) | 2459 (116) | 5299 (380) | 1401 (75) | 384 (22) | 4273 (152) | 3498 (271) | 78 (3) |

**Fig. 4** P$_{JEx}$-mediated transcription in non-enteric proteobacteria. **a** Fluorescence of *P. putida* KT2440, *S. (Ensifer) meliloti* Rm1021 and *C. crescentus* NA1000 cultures expressing RFP under the control of EilR-regulated P$_{JEx}$ promoters, either in their repressed state, or induced by 1 μM CV. The TetR-regulated P$_{LtetO-1}$ promoter[11], induced with 100 nM anhydrotetracycline, was included for comparison. **b** Fluorescence of cultures expressing RFP from three P$_{JEx}$ promoters induced by increasing CV concentrations. Data represent the averages of plate reader measurements from two (*P. putida*) or three (*S. meliloti, C. crescentus*) independently grown stationary phase cultures, with the standard deviation indicated by error bars (curves) or in parentheses (table). Fluorescence values are in arbitrary units and cannot be compared directly across different hosts because different instruments were used for measurements

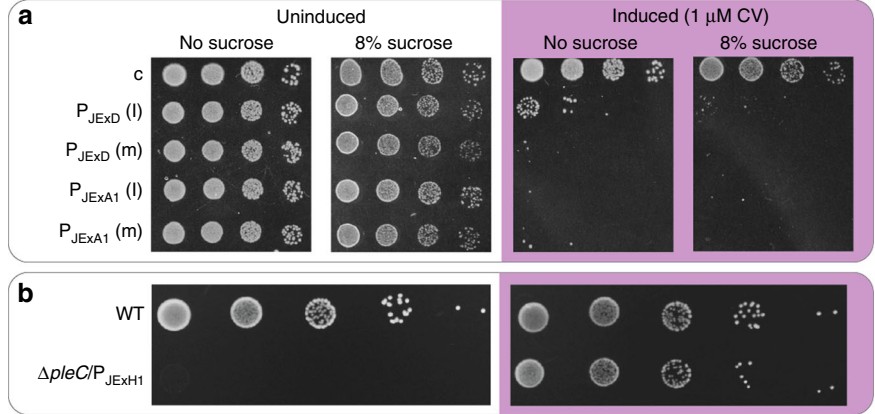

**Fig. 5** Tight regulation enables control of toxic and essential genes in *E. coli* and *S. meliloti*. **a** Ten-fold serial dilutions of *E. coli* expressing the toxic gene *sacB* from P$_{JExA1}$ and P$_{JExD}$ located either on low (l) or medium (m) copy-number plasmids, grown under repressed conditions or induced by 1 μM CV. A strain lacking *sacB* (c) was used as control. Induction caused *sacB* expression at levels that were toxic in the absence of sucrose or in the presence of high sucrose concentrations. Note that unlike in *sacB* counterselection plasmids, the P$_{JEx}$ plasmids do not include the native *B. subtilis* 5′ region, which contains a terminator element that is neutralized by a sucrose-inducible antiterminator[72]. **b** Ten-fold serial dilutions of either *S. meliloti* wild type (WT) or a Δ*pleC* mutant with a plasmid-borne sole copy of the essential cell cycle regulator gene *pleC* driven by P$_{JExH1}$ (Δ*pleC*/P$_{JExH1}$). Strains were plated under repressed conditions or induced by 1 μM CV

**Tight promoters for controlling bacterial phenotypes**. To demonstrate tight repressability of P$_{JEx}$ promoters, we first placed the toxic *sacB* gene under control of P$_{JExA1}$ and P$_{JExD}$ at medium and low copy-numbers in *E. coli*. This *B. subtilis* gene, encoding levansucrase, together with its upstream regulatory region are used as a counterselection marker in Gram-negative bacteria due to its conditional toxicity in the presence of sucrose. Under repressed conditions, the transformed *E. coli* grew normally in the absence or presence of high levels of sucrose. We observed that induction by 1 μM CV caused *sacB* expression at levels that are toxic in the presence, and unlike in conventional *sacB*-counterselection cassettes, also in the absence of sucrose (Fig. 5a).

Next, we showed that the P$_{JEx}$ system allows tight gene regulation in *S. meliloti*. We placed the cell cycle regulator gene *pleC*[39] under the control of P$_{JExH1}$ in an *S. meliloti* mutant lacking this essential factor. P$_{JExH1}$-mediated *pleC* repression caused a complete block of viability, while induction with 1 μM CV established normal growth (Fig. 5b). The ability to control the phenotype by tight gene regulation makes Jungle Express a useful instrument for physiological studies, bioengineering projects, and the expression of toxic proteins.

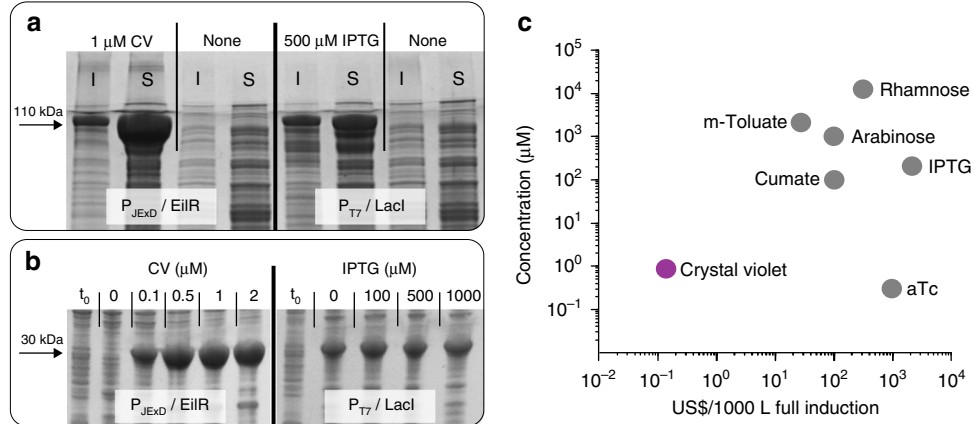

**Fig. 6** Protein overexpression from the EilR-mediated $P_{JExD}$ promoter in *E. coli*. **a** Production of the ~115 kDa tetrameric β-galactosidase protein LacZ expressed in *E. coli* BL21(DE3) from either the CV-inducible $P_{JExD}$ or the IPTG-inducible $P_{T7}$[53] on medium-copy plasmids (p15A *ori*). Soluble (S) and insoluble (I) fractions were separated prior to loading the SDS gel. **b** Production of the ~25 kDa RFP protein expressed from either the CV-inducible $P_{JExD}$ or the IPTG-inducible $P_{T7}$[53] on low-copy plasmids (pSC101 *ori*). Apart from the promoter/regulator cassette, the plasmids were identical. Crude cell extracts for SDS-PAGE were taken 4 h after induction and growth at 37 °C in LB-medium (**a**), or at the point of induction at early logarithmic phase ($t_0$) and 24 h after induction and growth at 30 °C in terrific broth (**b**). **c** Concentrations and costs of chemicals used to fully induce existing expression systems and EilR-regulated promoters in a 1000-L *E. coli* fermentation, based on extrapolation of lab-scale experiments

**Jungle Express provides high protein levels at low cost**. We applied Jungle Express as tool for protein production in an *E. coli* expression strain, and compared its expression capability with that $P_{T7}$, a strong promoter routinely used for high-level expression[10,35]. Inducing $P_{JExD}$ with 1 μM CV generated soluble and active β-galactosidase (Fig. 6a) at approximately three times higher levels than when expressed from the common, IPTG-inducible $P_{T7}$ (see Methods). Similarly, induction of $P_{JExD}$ with nM concentrations of CV enabled high production of the RFP protein at low gene copy numbers (Fig. 6b, Supplementary Fig. 5), with yields exceeding those of $P_{T7}$. While the T7-system displayed a high level of RFP expression in uninduced cells, no visible accumulation of the target protein was observed in repressed $P_{JExD}$ cultures.

While several of the EilR-inducing dyes, notably methylene blue and malachite green, inhibited growth of *E. coli* at high concentrations, CV, acridine orange, and pyronin had only minimal effects on growth rate at concentrations required for full induction (Supplementary Fig. 6). Minor mutagenic activities have been reported in different organisms for some of the dyes at higher concentrations[40]; however, the mutagenic effect of the optimal inducer CV at fully inducing levels (0.5 to 1 μM CV) was negligible in the hosts tested, *E. coli* and *S. meliloti* (Supplementary Table 1). These results demonstrate that the engineered promoters achieve high-level gene expression when induced by low concentrations of CV, a low-cost compound (Fig. 6c, Supplementary Table 2) that is stable in growth medium (Supplementary Fig. 7) and has marginal perturbation to the host.

**Mechanisms of operator recognition and ligand binding**. To characterize the mechanism of this system at the molecular level, we carried out structural and mutagenesis experiments. Using X-ray crystallographic analysis (Supplementary Table 3), we determined the structure of the repressor in complex with the consensus operator DNA, and found that EilR binds as a homodimer (Fig. 7a). Amino acid residues 1–192 of each monomer form 9 α-helices arranged in two domains: the N-terminal DNA-binding domain consists of residues 1–52, forming three α-helices, and the C-terminal ligand-binding domain

comprises residues 53–192, forming six α-helices. The C-terminal domain, like homologous domains in other members of the TetR family of transcription factors[41], is also responsible for dimerization. EMSA experiments confirmed that EilR functions as a single homodimer (Fig. 1b), similarly to TetR[41]. In contrast, QacR, another repressor of the TetR family, binds to its target DNA through two homodimers[42]. Similarly, the TetR family repressor RamR binds CV, but superpositioning the EilR and RamR structures in complex with CV indicated that the location of the CV binding site and the surrounding protein structure is significantly different[32] (Supplementary Fig. 8).

The DNA-binding domain of EilR contains a helix–turn–helix (HTH) motif, a common structure among DNA-binding proteins[43]. The HTH structure consists of two approximately perpendicular α-helices (α-helix2 and α-helix3) connected by a short turn. While EilR establishes non-specific DNA contacts through hydrogen bonding with the DNA sugar-phosphate backbone and a salt bridge with a phosphate group, three residues are responsible for the specific binding of EilR to its DNA operator (Fig. 7a, b): Residues Arg32 and His47 in the HTH domain establish direct contacts with two nucleotide bases in the *eilO* major groove. Unlike other characterized members of the TetR family[42], EilR also specifically interacts with a base located in the minor groove, established by residue Tyr3. To confirm the mechanism for specific EilR-*eilO* interaction observed by crystallographic analysis, we compared operator binding strength of wild type EilR with mutants containing Ala substitutions for amino acids Arg32, His47, and Tyr3 using EMSA binding assays (Supplementary Fig. 9). Arg32 and His47 substitutions completely impaired operator binding, while the decreased operator affinity of the Tyr3 mutant indicates that this interaction in the minor groove is important for binding specificity[44].

In a similar manner to the EilR-operator complex, we used crystallography to investigate the EilR structure in complex with each of two cationic triarylmethane ligands, malachite green (MG) and the most potent inducer CV (Fig. 2). Structural analysis revealed the binding of two ligand molecules per EilR dimer. This is similar to the TetR repressor in complex with tetracycline[41], but different from QacR, which binds only one ligand per dimer[45]. The two cationic triarylmethane ligands

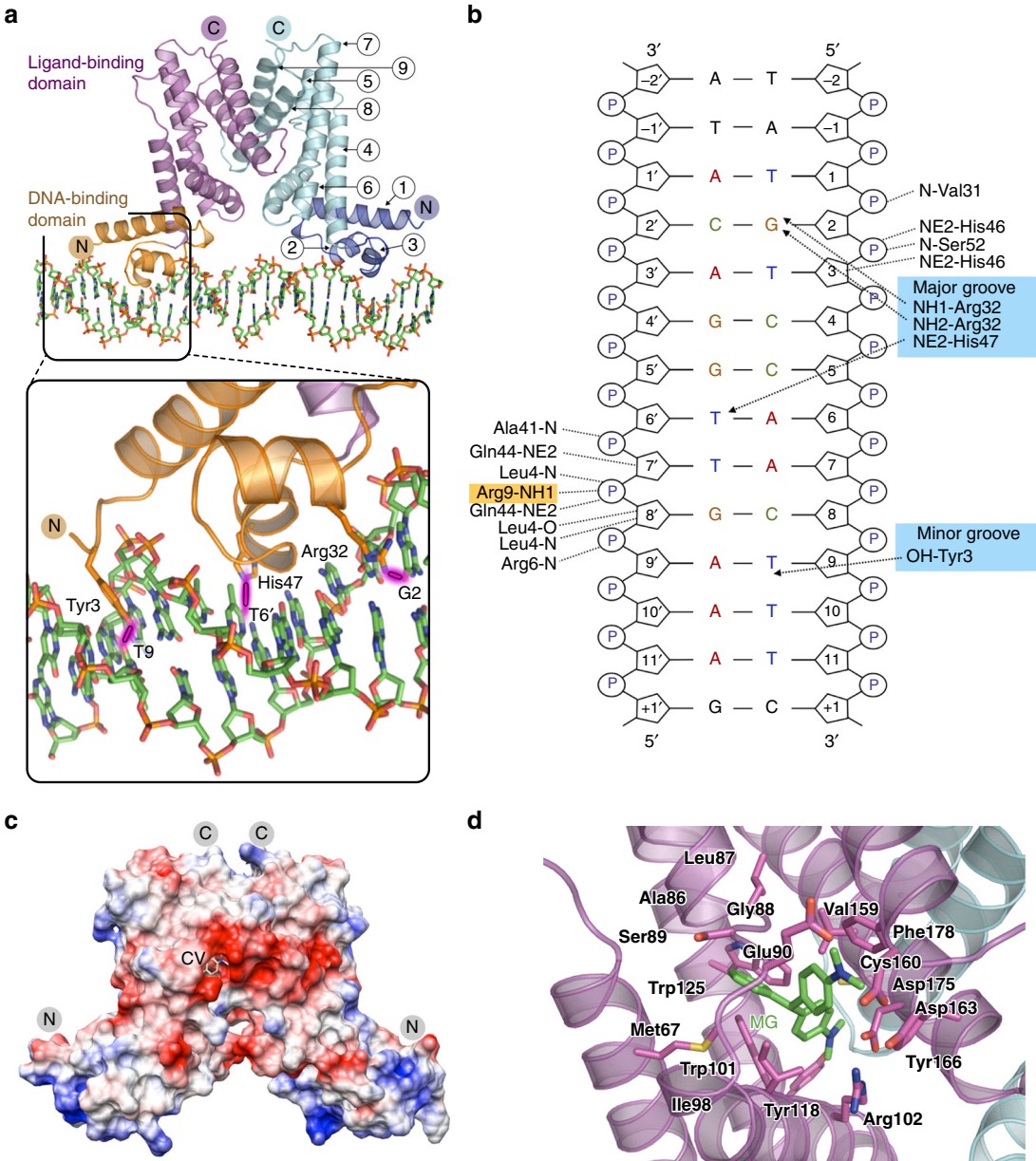

**Fig. 7** The structures of the EilR-*eilO* and EilR-CV complexes reveal direct interactions of EilR residues with nucleotide bases and inducing ligands. **a** Structure of EilR with α-helices 1–9 bound to its operator DNA. The inset shows direct contacts of EilR-residues with nucleotide bases via hydrogen bonds: The phenolic side chain of Tyr3 interacts with the pyrimidine moiety of $T^9$ located in the minor groove. The basic side chains of Arg32 and His47 from the helix-turn-helix region of the DNA-binding domain contact nucleotides $G^2$ and $T^{6'}$ in the major groove. **b** Schematic representation of hydrogen bonds observed between the EilR monomer and a duplexed 14-bp DNA containing the 11-bp half of the palindromic consensus *eilO* sequence (5′-$T^1G^2T^3C^4C^5A^6A^7C^8T^9T^{10}T^{11}$-3′). These contacts are responsible for operator-specific base recognition (blue background) and nonspecific interaction with the phospho-deoxyribose backbone. Arg9 (orange background) additionally builds a salt-bridge with a phosphate group. **c** An electrostatic surface representation of the repressor in complex with CV illustrates the strong negatively charged region around the ligand-binding pocket (red) and positively charged residues located in the DNA-binding domain (blue). Note that only one CV molecule is visible in this image. **d** EilR-binding pocket showing residues involved in electrostatic and van der Waals interactions with the inducer MG, which contacts residues Met67 in α-helix4; Ala86, Leu87, Gly88, Ser89, and Glu90 in α-helix5; Ile98, Trp101, and Arg102 in α-helix6; Trp125 in α-helix7; Val159, Asp163, and Tyr166 in α-helix8; and Phe178 in α-helix9

with a propeller-like geometry[46] bind within the negatively charged core of the C-terminal domain (Fig. 7c). The ligand-binding site of EilR contains several Glu and Asp residues (Fig. 7d), a distinctive feature this repressor shares with other transcription factors that recognize small cationic molecules[45,47]. To demonstrate the specificity of these negatively charged residues for ligand binding, we created Ala substitutions of EilR and examined their ability to de-repress RFP expression via $P_{JExD}$

in the presence of MG and CV (Supplementary Fig. 10). While the Glu90 and Asp175 mutants induced RFP to a similar extent as wild-type EilR, the absence of Asp163 caused a dramatic decrease in expression, suggesting that Asp163 is required for EilR to recognize the positive charge of these inducers.

Both CV and MG established hydrophobic interactions with the EilR by binding 14 residues from all six α-helices present in the C-terminal domain within van der Waals contact distance,

but CV established five additional van der Waals contacts via the extra dimethyl amino group (Fig. 7d, Supplementary Fig. 10), explaining the higher potency of CV compared to MG.

Comparison of EilR in its induced and DNA-bound forms showed a significant conformational change around the ligand-binding sites (Supplementary Fig. 11). A small difference in the positions of the DNA-binding domains of the EilR dimer was observed between the induced and DNA-bound states. Such a shift suggests a possible cascade event for the signal to be transferred to the DNA-binding domain, as has been previously been postulated for the TetR-repressor[48].

## Discussion

This work describes the development of an inducible broad-host expression system from scratch, taking an approach that can serve as a guideline for designing regulatable prokaryotic promoters. Out of the inexhaustible resource for genetic parts found in microbes, we made use of the multidrug efflux regulator EilR from a rainforest bacterium as the key component of this system. This repressor not only recognizes the major groove of its operator DNA like other members of the TetR-family, but it also establishes additional interactions with a base in the minor groove that result in increased DNA affinity and specificity. Phylogenetic analysis provided the second component required for tight regulation, a consensus operator with a higher affinity for EilR compared to that of the two native repressor-binding sites. Using a promoter-probe, we identified the third set of players, a range of inducing cationic dyes, of which the EilR-ligand CV acts with an $EC_{50}$ at nM concentrations. As a triarylmethane cation, CV is attracted by the negatively charged core of EilR, with Asp163 specifically required for tight binding, and its hydrophobic character increases affinity to the repressor via multiple van der Waals contacts. These additional contacts enhance the conformational changes within the EilR-dimer, which explains the positive cooperativity responsible for the high Hill coefficient in the concentration-response curve of CV.

Combining the fully symmetrical operator with immediate-early E. coli-phage promoters generated Jungle Express, an orthogonal and highly repressible bacterial expression suite with basal transcription levels falling below those of the most tightly known bacterial promoters. Activity of these engineered promoters increases up to 50,000-fold upon induction, enabling protein yields that exceed those obtained by one of the strongest existing bacterial expression systems. The high protein levels achieved by inducing at low gene copy numbers indicates that this system is also suitable for efficient expression from a single chromosomal copy, therefore permitting the development of robust and stable strains with reduced cellular burden and instability that would otherwise emerge from plasmid-based platforms[3,5]. While Jungle Express was designed for functionality in E. coli, it also efficiently regulates gene expression in three phylogenetically distant proteobacteria, including P. putida, an industrially relevant production host due to its metabolic versatility, low nutritional requirements, and high stress-tolerance[36,49]. EilR-based regulation is easily portable without requiring additional strain engineering, therefore facilitating the determination of a suitable organism for a target application.

Reagent choice is usually not an issue for lab-scale experiments, but in large-scale fermentations, media options are restricted and the costs of inducer compounds can become prohibitive, preventing maximal productivities. EilR-regulated promoters are media-independent and their induction is several orders of magnitude less expensive than that of existing systems, enabling the transition from culture tubes to industrial sized

bioreactors without tradeoffs between costs and performance. Jungle Express therefore serves as a scalable instrument for tightly regulated, high-level gene expression in a wide range of applications.

## Methods

**Reagents**. All chemicals were purchased from Sigma-Aldrich unless otherwise specified; plasmids are listed in Supplementary Table 4; and oligonucleotide primers in Supplementary Table 5.

**Preparation of dye stock solutions**. Aqueous solutions were prepared for all dyes except for Victoria Blue R, which was dissolved in 20% ethanol. Concentrations for all dyes were 10 mM, except for CV, for which we prepared a 1 mM stock solution.

**Phylogenetic analysis and design of a consensus operator**. The intergenic regions of eilR and eilA homologs in gamma-proteobacteria were extracted using pre-computed gene trees available in MicrobesOnline[50]. To improve specificity of motif reconstruction, we filtered out intergenic regions with more than 90% of sequence similarity using Jalview[51], which resulted in a set of non-redundant intergenic regions from the following bacteria: Enterobacter lignolyticus; Citrobacter koseri; Citrobacter rodentium; Salmonella enterica paratyphi; Salmonella enterica arizonae; Klebsiella pneumoniae 342; Klebsiella pneumoniae NTUH-K2044; Enterobacter sp. 638; Pantoea ananatis LMG 20103; Acinetobacter sp. ADP1; Acinetobacter baumannii. Intergenic regions from these organisms were used to identify putative EilR binding site motifs by MEME[52]. The MEME algorithm was applied with default parameters, restricting the motifs types to palindromes only and searching any number of site repetition on the same strand. The motif with the lowest E-value was considered as a putative eil-operator.

**Plasmid construction for E. coli assays**. All plasmids are listed in Supplementary Table 4. To construct the $P_{EilO1t}$ sensor strain, the eilR gene was PCR amplified from a fosmid containing E. lignolyticus genomic DNA that confers ionic liquid tolerance[27]. The eilR gene was then cloned after the weak constitutive promoter a$P_{FAB254}$ on pFAB5088 (provided by Vivek Mutalik), containing genes encoding kanamycin resistance and a monomeric red fluorescence protein (RFP) as reporter[12]. The resulting plasmid, pFAB_eilR was then used as template to generate the library of randomized −10 and −35 regions upstream of rfp. Primers were designed in a way to fit a truncated consensus eil-operator into a 17-bp spacer region between the −35 and −10 sites (see Supplementary Fig. 1). To create the randomized promoter library, pFAB_eilR was PCR amplified with the primers, eilO-pFAB_random_for and eilO-pFAB_random_rev (Supplementary Table 5), digested with DpnI (Thermo-Fisher), phosphorylated (using 100 ng PCR product) with polynucleotide kinase (Thermo-Fisher) and self-ligated with T4 DNA ligase (Thermo-Fisher) at 16 °C for ca. 14 h. One microliter purified ligation product was transformed into chemically competent E. coli DH10B containing plasmid pBbS5c-eilA (SC101 ori, Cm$^R$) enabling [C$_2$C$_1$im]Cl-tolerance via the IPTG-inducible eilA gene. Transformed cells were plated on 200 × 200 mm LB agar plates supplemented with kanamycin (50 mg L$^{-1}$) and chloramphenicol (12.5 mg L$^{-1}$) and incubated at 37 °C overnight. One hundred and thirty six colonies were transferred into 96-deep-well microtiter plates and grown to stationary phase in EZ-Rich media containing 0.2% glucose and 10 μM IPTG either without or with 300 mM [C$_2$C$_1$im]Cl. To identify variants that respond to [C$_2$C$_1$im]Cl, RFP fluorescence of cells was measured in a Tecan F200pro plate reader. Promoter $P_{EilO1t}$, located on the resulting plasmid pFABeilO1t, is the variant with the highest dynamic range.

After removal of the BglII-site upstream of $P_{EilO1t}$, the region spanning from eilR to the transcriptional start downstream of $P_{EilO1t}$ was transferred from pFAB_eilR between the AatII and the EcoRI sites of a BglBrick plasmid backbone[53] (p15A ori, Kan$^R$) containing the rfp gene and its RBS by isothermal DNA assembly, following the manufacturer's instructions (New England Biolabs), which resulted in plasmid pTR_EilO1t.

To construct the $P_{JEx}$-promoter suite, phage promoters $P_{A1}$, $P_{H207}$, $P_{DE20}$, and $P_L$[29] with truncated eilO operators in their 17-bp spacer regions were ordered as gBlocks (IDT), with the flanking regions containing at least 40-bp identity with ends of the PCR-amplified modified version of pFAB_eilR. gBlocks were cloned into the linearized vector backbone by isothermal DNA assembly. The resulting plasmids were PCR-amplified with primers that each contained half an operator at the transcriptional start (Supplementary Table 5). PCR products were self-ligated to obtain promoters with two eilO operators. $P_{JEx1}$ was generated by taking the same approach, using pTR_EilO1t as template plasmid. All assemblies were transformed into E. coli DH10B, and the promoter region and rfp sequence-verified.

To engineer sacB-plasmids, the sacB gene was PCR-amplified from pKW1 (sacB counterselection suicide plasmid, gift from Kelly Wetmore) to replace the rfp gene on pTR_sJExA1-rfp, pTR_aJExA1-rfp, pTR_sJExD-rfp, and pTR_aJExD-rfp via Golden Gate cloning, while the RBS on these plasmids was maintained. The resulting plasmids pTR_sJExA1-sacB, pTR_aJExA1-sacB, pTR_sJExD-sacB, and pTR_aJExD-sacB were transformed into E. coli DH10B and sequence-verified.

To generate *lacZ*-plasmids, the *lacZ* gene was PCR-amplified from *E. coli* MG1655 genomic DNA to replace the *rfp* gene on pTR_aJExD-rfp and pBbA7k-rfp via Golden Gate cloning, while the RBS on these plasmids was maintained. Plasmids were transformed into *E. coli* DH10B and sequence-verified.

**E. coli fluorescence measurements**. Cells were induced for RFP expression as indicated and measured after growing at 37 °C to stationary phase, unless otherwise described. Microplate measurements were performed in a BioTek Synergy 4 reader for absorbance at 600 nm and fluorescence (575 nm excitation, 620 nm emission).

**Flow cytometry**. Single-cell fluorescence and population homogeneity were measured in stationary phase *E. coli* cultures expressing RFP after a 1:200 dilution in PBS buffer. An LSRII Fortessa (BD, CA, USA) instrument, equipped with a yellow–green laser (561 nm excitation) was used to detect mRFP fluorescence during dynamic range measurements shown in Fig. 3a, b. For each sample, 50,000 events were measured with the following settings: FSC-H (forward scatter): 473 V, SSC-H (side scatter): 279 V, PE-Texas Red-H: 450 V (mRFP detection). A Guava easyCyte (Millipore) flow cytometer was used for generating the histogram in Fig. 3c. For each sample, 5000 events were counted by forward and side scatter acquisition, and the cellular accumulation of RFP was measured by fluorescence intensity. Data acquisition was performed using InCyte software version 2.2 (Millipore).

**EilR-regulated promoters in other bacteria**. Maps of plasmids constructed for assays in *P. putida*, *S. meliloti* and *C. crescentus* are shown in Supplementary Fig. 12. The broad-host-range vector pJC543 was assembled using In-Fusion HD Cloning Kit (Clontech), by inserting *tetR* from pZS4Int1[11] into pZE21-MCS1 at the *Bgl*II site, and the RK2-based origins of replication and conjugative transfer (*oriV*-*oriT-trfA*) from pCM130[54] at the *Spe*I site. Specifically, *tetR* was amplified using primers DVA00311 (5′-ACGATCCTCATCCTGTCTCTTGATCACGATCGTTA AGACCCACTTTCACATTTAAGTTG) and DVA00312 (5′-AAGGATCTGATGG CGCAGGGGATCAAGATCTATGTCTAGATTAGATAAAAGTAAAGTGA), while *oriV-oriT-trfA* was amplified using primers DVA00309 (5′-CTCACGTTAA GGGATTTTGGTCATGAACTAGTCTAGCGTTTGCAATGCACCAGG) and DVA00310 (5′-GGGGCGTTTTTTATTGGTGAGAATCCAAGCAGCTAGCCTGC CATTTTTGGGGTGAGGCCG).

The two PCR products were combined with the two fragments of pZE21-MCS1 resulting from digestion with *Bgl*II and *Spe*I. The assembled pJC543 plasmid, which contains the P_LtetO-1 promoter and encodes its cognate regulator TetR, can be conjugated into a wide range of species by selecting for resistance to kanamycin or neomycin.

Next, the *Eco*RI-*Avr*II fragment containing *rfp* (iGEM part BBa_E1010) and the *dbl* transcriptional terminators (part BBa_B0015) from pBbB2k-RFP[53] was inserted into pJC543, using the same restriction sites downstream of P_LtetO-1, thus replacing the *rrnB* T1 transcriptional terminator and generating pJC548. Expression of RFP from this plasmid can be induced with anhydrotetracycline.

The first series of plasmids (pJC566—pJC573) containing EilR-regulated promoters (P_JEx1, P_JExA1, P_JExA2, P_JExL, P_JExD, P_JExH1, and P_JExH2, respectively) was constructed by inserting each promoter and the divergently transcribed *eilR* into pJC548 to replace P_LtetO-1 and *tetR*, using In-Fusion HD. Promoters and *eilR* were amplified from the pTR_aJEx plasmid suite using primers Kan-R-ig-F (5′-GCGA AACGATCCTCATCCTG) and RFP-R (5′-GTCTTCGCTACTCGCCATATG), and the PCR products were each combined with the desired *Pvu*I-*Eco*RI fragment from pJC548. This series of plasmids could be introduced into *P. putida* KT2440 and *S. meliloti* Rm1021 but not *C. crescentus* NA1000. Repeated attempts led to recovery of plasmids with transposon insertions between *nptII* and *oriV*, suggesting possible interference of plasmid replication due to transcription read-through from *eilR* and *nptII*.

The EilR-regulated expression plasmids were refined by inverting the region containing the origins of replication and transfer and inserting the *V. fischerii luxG* terminator (iGEM part BBa_B0011) downstream of *trfA*, to generate pJC575—pJC582 (containing P_JEx1, P_JExA1, P_JExA2, P_JExL, P_JExD, P_JExH1, and P_JExH2, respectively). For example, to construct pJC575, the *oriV-trfA* region was amplified from pJC566 with primers DVA00608 (5′-CACGTTAAGGGATTTTGGTCATG AACTAGTCTGCCATTTTTGGGGTGAGGCCG) and DVA00609 (5′-CGGGCG TTTTTTATTGGTGAGAATCCAAGCAAATAATAAAAAAGCCGGATTAAT AAT CTGGCTTTTTATATTCTCTGCTAGCGTTTGCAATGCACCAGG), and the resulting PCR product was combined with the appropriate *Nhe*I-*Spe*I fragment from pJC566. For comparison, a TetR-regulated expression plasmid pJC583 was constructed by ligating the *Eco*RI-*Sac*I fragment containing *rfp*, *colE1*, *oriV*, *oriT*, and *trfA* from pJC575 and the *Eco*RI-*Sac*I fragment containing *nptII*, *tetR*, and P_LtetO-1 from pJC548. A plasmid without *rfp*, pJC586, which served as the vector control, was constructed by ligating the *Sph*I-*Avr*II fragment containing *tetR*, P_LtetO-1, and *rrnB* T1 from pJC543 with the *Avr*II-*Sph*I fragment containing *colE1*, *oriV*, *oriT*, and *trfA* from pJC575.

Plasmid pJC681 was constructed by amplifying SMc02369 from pJC476 with primers pleC −20F BamHI (5′-GCGGGATCCCAGGACGACAAATTGGATAAG) and SMc02369+17R BamHI (5′-TTAGGATCCAGCGTAAGCGGGCGTGGT CA), digested with *Bam*HI, and inserted into pJC581, digested with *Bgl*II and *Bam*HI to replace *rfp*. Sequences of all fragments amplified by PCR were verified in

the resultant plasmids. DNA parts used for a subset of the plasmid construction were designed using DeviceEditor and j5 software tools[55] and assembled via isothermal DNA assembly.

**Growth and induction of non-E. coli strains**. Plasmids were maintained in *E. coli* DH10B (Invitrogen), which was cultured using lysogeny broth (LB) supplemented with 30 (in liquid medium) or 50 (in solid medium) µg ml⁻¹ kanamycin. *C. crescentus* NA1000[56] and *S. meliloti* Rm1021[57] were grown in peptone-yeast extract (PYE) medium and *P. putida* KT2440[58] in LB medium, with antibiotics when appropriate: chloramphenicol (12.5 µg ml⁻¹ for KT2440), kanamycin (5 (liquid) or 25 (solid) µg ml⁻¹ for NA1000; 25 µg ml⁻¹ for Rm1021; 50 µg ml⁻¹ for KT2440), nalidixic acid (20 µg ml⁻¹ for NA1000 and Rm1021), and neomycin (50 µg ml⁻¹ for Rm1021).

Mobilization of plasmids from *E. coli* DH10B to *S. meliloti* or *C. crescentus* was accomplished by triparental mating, with the help of strain MT616, carrying pRK600[59] (conferring resistance to chloramphenicol); nalidixic acid was used to select against *E. coli* donor and helper strains. Similarly, *E. coli* strain HB101/ pRK2073[60] (conferring resistance to spectinomycin) facilitated conjugation into *P. putida*; chloramphenicol was used to select against the donor and helper strains.

RFP expression was monitored with 160 µL cultures grown in 96-well plates (Corning Falcon 353072) at 30 °C for up to 24 h. For *P. putida* KT2440-derived strains, overnight cultures were diluted to an optical density at 600 nm of 0.015, and 80 µL aliquots of the diluted cultures were dispensed into each well containing 80 µL of LB medium containing kanamycin, with varying concentrations of anhydrotetracycline or crystal violet. Absorbance at 600 nm and fluorescence (575 nm excitation, 620 nm emission) were measured every 20 min in a Tecan Infinite F200 PRO plate reader. For *C. crescentus* NA1000, overnight cultures were diluted to an initial optical density at 600 nm of 0.1 in PYE medium containing kanamycin for distribution into wells, and absorbance at 590 nm and fluorescence (535 nm excitation, 620 nm emission) were measured in a Tecan Infinite F200 plate reader. For *S. meliloti* Rm1021, overnight cultures were diluted to an initial optical density at 600 nm of 0.2 in PYE medium containing kanamycin, and absorbance at 600 nm and fluorescence (575 nm excitation, 620 nm emission) were measured in a BioTek Synergy 4 plate reader.

**Spotting assays**. *E. coli* DH10B containing pTR_sJExA1-sacB, pTR_aJExA1-sacB, pTR_sJExD-sacB, pTR_aJExD-sacB or the non-*sacB* control plasmid pBbA0k were grown overnight in LB/Kan (50 µg/mL) and diluted to an OD600nm = 1 for 10-fold serial dilutions. Three microliter were spotted on LB/Kan (50 µg/mL) supplemented with 8% sucrose and/or 1 µM CV, and colonies from OD600nm 10⁻² to 10⁻⁵ dilutions were photographed after 20 h growth at 37 °C.

The *S. meliloti pleC* gene under the control of P_JExH1 (pJC681) was introduced into strain JOE3608[39] [Δ*pleC*::Ω/pJC476 (P_tau-*pleC*)] by triparental mating, selecting for neomycin resistance in the presence of 100 mM taurine and 1 µM CV, to replace the complementing plasmid pJC476, resulting in strain JOE5635. Stationary-phase cultures of JOE5635, JOE3608 (Δ*pleC*/P_tau-*pleC*) and JOE3593 [Rm1021 (*pleC*⁺)/P_tau-*pleC*] were washed and resuspended in PYE to an optical density at 600 nm (A600) of 0.1, and serially diluted 10-fold in water. Five microliter of each dilution (from 10⁻² to 10⁻⁶) was spotted onto PYE plates, without or with 100 mM taurine or 1 µM CV, and incubated at 30 °C for 4 days prior to imaging.

**SDS-PAGE**. For RFP expression, low-copy-number RFP expression plasmids pBbS7k-RFP (LacI/P_T7) and pTR_sJExD-rfp (EilR/P_JExD) were transformed into BL21-(DE3), which were then grown in 50 mL TB+Kan (50 µg/mL) to an OD600 of ~1.2. Five ml of each culture was transferred to 25 mL glass culture tubes. One sample of each strain was stored at −20 °C as un-induced sample. The other cultures were induced with the indicated amount of CV or IPTG and grown at 30 °C for 24 h at 200 rpm. Samples were then centrifuged, re-suspended in 50 µL of 1× SDS-PAGE sample buffer (SB) and heated for 6 min in the microwave. Samples (1 µL) and a Novex sharp prestained protein standard were loaded onto 8–16% gels.

For β-galactosidase (LacZ) expression with either T7 or P_JEx promoters, BL21-DE3 cells were transformed with the medium-copy-number plasmids pBbA7k-lacZ (LacI/P_T7) and pTR_aJExD-lacZ (EilR/P_JExD). Cells were grown in 5 mL LB + Kan (50 µg/mL) in glass culture tubes at 37 °C until an OD600nm of ~1.0. The cultures were then induced with either 500 µM IPTG or 1 µM CV and grown at 37 °C for 4 h. OD7 pellets (~1 ml culture at OD600nm of 7) were collected and stored at −20 °C. An uninduced culture for each plasmid was also grown and sampled. The pellets were thawed, re-suspended in 700 µL PBS with 3 µg/mL DNase I. The cells were lysed by sonication, and the lysate was centrifuged to separate the soluble from insoluble fractions. The insoluble fraction was resuspended in 700 µL PBS and 7 µg of both the soluble and insoluble samples were analyzed by SDS-PAGE.

**β-Galactosidase (LacZ) activity measurement**. Activity of β-galactosidase was measured in the soluble fraction of the lysates used for the SDS-PAGE analysis, following an established spectrophotometric assay using ONPG (ortho-nitrophe-nyl-β-galactoside) as the substrate[61]. This involved mixing 1 µL of lysate with 276 µL 0.1 M Na-phosphate (pH 7.5) buffer containing 20 µL ONPG (4 mg/mL in 0.1 M Na-phosphate buffer), 3 µL 0.1 M MgCl₂, and 4.9 M β-mercaptoethanol.

Mixtures were incubated for 15 s at room temperature, prior to stopping the reaction with 600 μL 1 M $Na_2CO_3$. Absorbance was read on a spectrophotometer at 420 nM, using $H_2O$ as blank. Readouts were 0.189 ($P_{T7}$/IPTG), 0.586 ($P_{JExD}$/1 μM CV), and 0.021 (sample without lysate).

**Purification of the EilR protein**. EilR-His$_8$ was expressed in *E. coli* harboring a pET-derived expression plasmid, pLane-eilR, with an IPTG-inducible T5 promoter and a TEV protease-cleavable *his8*-tag. Cells were grown to stationary phase overnight, and diluted 1:100 in 500 mL Terrific Broth (TB) supplemented with 2 mM MgSO$_4$ for cultivation in 2-L non-baffled flasks. These cultures were grown at 37 °C shaking at 200 r.p.m. until the OD$_{600}$ was ~1.3, then the temperature was lowered to 20 °C, IPTG was added to 0.5 mM, and the cultures continued for 3 days. For crystallography, selenomethionine (Se-Met) labeled protein was produced using the method described by Studier[62]. At this time the cells were harvested and the pellets stored at −80 °C. Expression levels were estimated using SDS-PAGE. Protein purification was begun by thawing the paste and re-suspending it in 50 mM Tris buffer, pH 8.0, containing 600 mM NaCl, 50 mM Na-glutamate, 50 mM arginine-HCl, 10 mM MgCl$_2$, and 0.5 mM dithiothreitol (high salt buffer, HSB). The re-suspended cells were lysed using the Emulsiflex C3 homogenizer. The lysate was clarified by centrifugation at 40,000×*g* for 40 m at 4 °C. The clarified lysate was loaded onto a 5 mL His-Trap column and fractionated using an AKTA FPLC. The column was washed with HSB to establish an OD$_{280}$ baseline prior to applying 100 mL (20 column volume) of a gradient of 2–99% of 1 M imidazole in HSB. To remove both imidazole and the His$_8$ tag, fractions containing EilR-His$_8$ were pooled and TEV protease was added at a 1:100 molar ratio. The pooled fraction was dialyzed against 1 L of HSB overnight at 4 °C. Cleavage was monitored by SDS-PAGE analysis of an aliquot. The dialysate was passed through a 1 mL His-Trap column to capture TEV and remaining His-tagged EilR. EilR lacking the His$_8$ tag was collected in the flowthrough. For crystallographic studies, purified EilR was dialyzed against 1 L of 50 mM Tris buffer, pH 8.0, containing 150 mM NaCl, 50 mM Na-glutamate, 50 mM Arginine-HCl, 10 mM MgCl$_2$, and 0.5 mM dithiothreitol, and concentrated to 10.5 mg mL$^{-1}$.

**Electrophoretic mobility shift assays (EMSA)**. In the assay comparing EilR affinity to various operator versions, the purified EilR molecules and the duplexed oligonucleotides (Supplementary Table 6) were mixed in 30 mM Tris, pH 7.7, 100 mM NaCl, 25 mM arginine, 25 mM glutamine, 5 mM MgCl$_2$ and left at room temperature for 1.5 h before their run in a 2% agarose gel in Tris-borate-EDTA buffer. A total of 231 pmoles of protein were used for the 2:1 EilR:DNA, and 462 pmoles of protein for the 4:1 EilR-DNA mixtures. The samples were run on a 2% agarose gel in TBE buffer stained with SYBR safe dye (Invitrogen), and imaged with an Alpha Innotech FluorChemQ instrument.

In the EilR-mutant EMSA assay, EilR versions were mixed with the duplexed oligonucleotide 5′-AA**AAAGTTGGACACGTGTCCAACTTT**CC-3′ (*eilOc* operator in bold letters) in 50 mM Tris, pH 8.0, 150 mM NaCl, 50 mM arginine, 50 mM glutamine, 10 mM MgCl$_2$, and 0.5 mM DTT and left at room temperature for 30 min before running in a 2% agarose gel in Tris-borate-EDTA buffer.

**Crystallization of EilR in complex with *eilOc* and inducers**. The final concentration of EilR used for crystallization trials was 10 mg mL$^{-1}$. Oligonucleotides were synthesized at the 1 μmol scale and purified to remove small-molecule impurities by commercial vendors, such as IDTDNA (Coralville, Iowa). The oligonucleotides were resuspended in 20 mM Tris–HCl pH 7.5 containing 10 mM MgCl$_2$. Oligonucleotide pairs were annealed in equimolar ratios by heating at 95 °C for 10 min and gradual cooling to room temperature. The EilR–*eilOc* complexes were formed by adding the duplexed oligonucleotide AA**AAAGTTGGA-CACGTGTCCAACTTT**CC-3′ (*eilOc* operator in bold letters) to the protein solution in a 2:1 (protein:DNA) molar ratio. The EilR apoenzyme and EilR-*eilO* complexes were screened using the sparse matrix method[63] with a Phoenix Robot (Art Robbins Instruments, Sunnyvale, CA) and the following crystallization screens: Berkeley Screen (Lawrence Berkeley National Laboratory, Berkeley, CA), Crystal Screen, SaltRx, PEG/Ion, Index, and PEGRx (Hampton Research, Aliso Viejo, CA). Crystals of EilR apoenzyme were formed in 0.2 M trisodium citrate and 20% (w/v) polyethylene glycol 3350. Crystals of EilR-MG and EilR-CV complexes were obtained by soaking the crystallized EilR apoenzyme in 1 mM MG or CV solution for 5 h. The EilR-*eilO* complex was crystallized in 0.1 M Li$_2$SO$_4$, 0.1 M MgCl$_2$, 0.1 M 4-(2-hydroxyethyl)-1-piperazineethanesulfonic acid (HEPES) buffer (pH 7.5), 20% (w/v) polyethylene glycol 3350 and 10% hexanediol. EilR apoenzyme and EilR-*eilO* crystals were obtained after 3 days by the sitting-drop vapor-diffusion method with the drops consisting of a mixture of 0.2 μL of protein solution and 0.2 μL of reservoir solution.

**X-ray data collection and structure determination**. The crystals of EilR-*eilO*, EilR-MG, and EilR-CV were placed in a reservoir solution containing 20% (v/v) glycerol, then flash-frozen in liquid nitrogen. X-ray data sets were collected at the Berkeley Center for Structural Biology beamlines 8.2.2 of the Advanced Light Source at Lawrence Berkeley National Laboratory (LBNL). The diffraction data were recorded using an ADSC-Q315r detector. The data sets were processed using the program HKL-2000[64]. The EilR-*eilO* complex structure was determined using selenomethionine-labeled protein by the single-wavelength anomalous dispersion method[65] with the *phenix.autosol*[66] and *phenix.autobuild*[67] programs. The EilR-MG and EilR-CV complex structures were determined by the molecular-replacement method with the program *PHASER*[55] taking the EilR structure from the EilR-*eilO* complex as the search model. Structure refinement was performed by *phenix.refine* program[68]. Manual rebuilding using COOT[69] and the addition of water molecules allowed for construction of the final model. Five percent of the data were randomly selected for cross validation. The final models of the EilR-*eilO*, EilR-MG, and EilR-CV complexes showed an R factor of 19.4%/R$_{free}$ of 22.8%, R factor of 19.6%/R$_{free}$ of 24.0% and R factor of 20.4%/R$_{free}$ of 27.1%, respectively. Root-mean-square deviations from ideal geometries for bond lengths, angles, and dihedrals were calculated with Phenix[70]. The overall stereochemical quality of the final models for EilR-*eilO*, EilR-MG, and EilR-CV were assessed using the Mol-Probity program[71].

## Data availability

All plasmids listed in Supplementary Table 4 have been deposited in the Joint BioEnergy Institute Public Registry[78] (https://public-registry.jbei.org/folders/378) and are available for searching and reviewing the sequences and annotations. Sequences for the *Enterobacter lignolyticus eilR* gene, *eilO* operator, promoter, and EilR protein have the GenBank accession number MH668001. The atomic coordinates and structural factors of EilR-*eilO*, EilR-MG, and EilR-CV complexes have been deposited in the Protein Data Bank: 5VL9, 5VLG, and 5VLM, respectively.

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

## Acknowledgements

TLR is grateful to Prof. Thomas Boller for supporting independent graduate research abroad. We thank Ms. Garima Goyal for assisting assembly of conjugative plasmids, Mr. Zhengzhong Huang for assistance with strain construction and mutant frequency analysis, and Prof. Adam Arkin for providing access to the flow cytometer. We also thank Prof. Urs Rüegg for insightful comments on the manuscript. This work was part of the DOE Joint BioEnergy Institute (http://www.jbei.org), supported by the U.S. Department of Energy, Office of Science, Office of Biological and Environmental Research, through contract DE-AC02-05CH11231 with Lawrence Berkeley National Laboratory and contract DE-AC52-07NA27344 with Lawrence Livermore National Laboratory. T.L.R. was funded by the Swiss National Science Foundation via a Doc.Mobility scholarship. J.C.C. was supported by the NIGMS under Award Number SC3GM096943 and by the CSU Program for Education and Research in Biotechnology. We are grateful to the staff of the Berkeley Center for Structural Biology at the Advanced Light Source, which is supported in part by the National Institutes of Health, National Institute of General Medical Sciences, and the Howard Hughes Medical

Institute. The Advanced Light Source is a Department of Energy Office of Science, Office of Basic Energy Sciences, User Facility under Contract No. DE-AC02-05CH11231.

## Author contributions

T.L.R. conceived and designed the experiments for promoter development and inducer identification. V.K.M. and T.L.R. conceived the sensor plasmid. T.L.R. performed promoter experiments in *E. coli* except for protein overexpression. J.H.P. crystallized EilR and elucidated its structure. J.C.C. performed experiments in non-enteric hosts. P.N. performed phylogenetic analysis. A.D. performed EMSA experiments and protein overexpression, and A.D. and G.P.T. purified EilR for crystallization. B.A.S., P.D.A., N.J.H., S.W.S., and J.C. C., contributed reagents and research facility infrastructure, and M.P.T. coordinated research activities leading to this publication. T.L.R., M.P.T., J.C.C., J.H.P., and V.K.M. wrote the manuscript. P.D.A. and N.J.H. reviewed the manuscript, and M.P.T. prepared the manuscript for publication. All authors read and approved the final manuscript.

## Additional information

**Competing interests:** T.L.R. at LBNL has applied for a patent (application US20170002363) for the development and applications of the Jungle Express system. The remaining authors declare no competing interests.

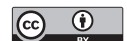

