## [Peer Review File · Nature Communications]

Reviewers' comments:

Reviewer #1 (Remarks to the Author):

The authors develop a bacterial expression system based on the previously identified repressor protein EilR. To this aim they identify a consensus operator sequence and clone this operator in various configurations into existing bacterial promoters. They characterize the performance of this promoter with regard to inducer spectrum (e.g. crystal violet, acridine orange), expression kinetics and dynamic range. They demonstrate that their promoter compares favorably with state of the art expression systems (arabinose system, TET-promoter, LacI). They demonstrate that their promoter performs well in different bacteria such as *P. putida*, *S. meliloti* or *C. crescentus*. They further solve the structure of EilR and perform mutation studies to identify the molecular basis for DNA-binding and inducer recognition.

The availability of strong bacterial expression systems that are responsive to cheap inducers is of high interest for biotech industry, especially when commodity proteins are to be produced where expensive inducers would negatively impact on the economic feasibility of the process.

In order to be able to better evaluate the potential of this approach but also to get to know potential limitations, it is recommended that the authors address the following points:

1. The favored inducer in the manuscript is crystal violet. Crystal violet is a well-known dye staining bacterial walls (Gram-staining). Furthermore, crystal violet is known to stain DNA and to be mutagenic (see MSDS of crystal violet, PMID 6472325, https://pubchem.ncbi.nlm.nih.gov/compound/Crystal_violet). Such mutagenic activity could induce mutations in the production strain including the target gene sequence and thus lead to product heterogeneity. The authors should experimentally analyze whether crystal violet at concentrations required for maximum gene expression (e.g. 2 μ M, according to Supplementary Fig. 4) induces mutations in the host strain. The authors should as well analyze the target gene for mutations after a typical time span required for protein production (e.g. up to 12 h).
2. Similar point: in Supplementary Figure 8 the authors demonstrate that the used inducers negatively impact cell growth (which is also in agreement with previous literature). This seems also the case for crystal violet at concentrations of 2 μ M (the concentration where maximum gene expression is obtained, see Supplementary Fig. 4). Is this toxic effect caused by DNA mutations? The authors should accordingly rephrase their statement on Fig. 5, last line.
3. The authors should perform a 1:1 comparison of their system to the most commonly used platforms for the production of recombinant proteins in everyday molecular biology and biotechnology. For example, they should compare their system to IPTG-inducible expression in *E.coli* BL21 (DE3) and *E.coli* BL21 (DE3) pLysS with regard to basal levels, expression kinetics as well as maximally induced levels.
4. What is the stability of crystal violet in culture medium? Is this stability dependent on bacterial density or do bacteria somehow absorb the dye (less likely for Gram-negative bacteria)? If there is an effect of bacterial density: how would this affect performance in routine high density fermentations in which bacterial densities more than 100x higher than reported here are used? Would it then be required to add higher crystal violet concentrations to compensate for the degradation/absorption? Would these concentrations be toxic?
5. Previously, the crystal structure of a similar repressor protein (RamR) has been reported that binds to similar or the same (e.g. crystal violet) inducers (Yamasaki et al. 2013, doi: 10.1038/ncomms3078). It would be interesting to compare this previous structure to the one of EilR in terms of inducer recognition and DNA-binding properties.

Minor point

6. The 3D representation of Supplementary Figs. 6 and 7 is difficult to interpret (some bars seem hidden behind others). The authors should use a 2D representation where the RFP values for every

single condition can clearly be seen. Also, the authors should indicate the number of replicates (for Supplementary Fig. 6) and indicate error bars in both Supplementary Figures.

Reviewer #2 (Remarks to the Author):

Overview:

In this paper, the authors claim that they have developed an inducible expression system in bacteria with features of tight control and strong expression level, as well as activity in distinct bacterial species, compared to commonly used inducible systems. Also, the authors emphasize the cost efficiency of the inducer for this expression system, which has the potential for industrial-sized fermentations. A 3-D structure of the repressor protein bound to ligand was solved and presented. Overall, the authors provided data supporting each point they claimed. However, the data were in some cases insufficient and poorly organized and presented. There was a general lack of detailed description and labeling, which caused confusion and raised questions.

What they claimed and what they still need to provide

1. Development of expression system and inducer screen

a) The authors phylogenetically derived a consensus operator sequence for the previously identified repressor EilR, and engineered a series of promoters based on this operator sequence. The biggest problem here is that there is no accession number provided for each operator sequence, nor there an entry for EilR in the protein (NCBI) database nor indeed for the Xray structure (pdb file). All these sequences must be deposited in the relevant databases to make this paper publishable.

b) Another concern here is the engineering process is poorly described. A flow chart is needed to illustrate the construction of the library, the subsequent screen, and the incorporation of Phage promoters. Figure 1c, Method, and Supplementary Figure 2, which are related to this section, are inconsistent.

c) Data from the library screen are missing. RFP expression data for all the colonies picked from the library should be provided.

2. Regulation evaluation: the authors claimed that EilR system is tight and strong, media-independent, population-uniform and host-versatile.

d) The authors have provided flow cytometry and immunoblot data for RFP expression under regulation of EilR system, indicating a low basal level and high expression with a relatively low concentration of inducer. However, the authors only tested the system with one gene, RFP, which is insufficient. At least one more gene should be tested to be convincing.

e) In addition to flow cytometry, more analysis approaches should be included for the expression level. At minimum some other unrelated protein (lacZ?) should be tested. That's a very easy assay, a substantially larger protein, and should also be evaluated by immunoblot.

f) The authors claimed the EilR system is media-independent, but failed to provide the analysis data for different culture media.

3. Comparison to common inducible promoters

Reviewer #3 (Remarks to the Author):

In this study, the authors describe the development of a new method to induce protein over-expression in bacteria. It is based on their previously discovered bacterial multidrug efflux system. Normally, this system controls the tolerance to environmental (toxic) compounds, where the compounds directly de-repress of the expression of membrane transporters. The authors identified the cognate repressor (EilR) DNA binding sequence, and use this to design and create a palindromic consensus operator that shows a strong and improved binding affinity to the repressor. By combining this DNA sequence with commonly used viral promoters (that are

recognized by the host polymerases), they created an expression system that allows high level protein production with precise time control, which they named Jungle Express (JEx).

The system does rely on the addition of natural toxic compounds to the medium. However, the sensitivity of this system allows the use of very low concentrations of cheap inducers that are tolerated by the cells. This is a key feature, because it would allow the induction of large scale cultures, something that is problematic with existing systems. Additionally, JEx does differentially respond to a range of compounds, allowing for tuning the induction for your needs.

Together this system promises high-level and tunable protein expression in bacteria, and adds a couple of features that set it apart from widely used analogous system. Overall, I believe the study is well executed and will find an audience in both research and industry.

Minor points:

- What are the respective concentration of the binding partners in gel shift assays (see for example Fig1b, S1 and S9)?
- What is the unit of the concentration of [C2C1mim]Cl in figure S2d?
- A potential drawback of the system is the use of toxic compounds. The authors do address this point in figure S8, where they concluded that a few compounds did not affect cell growth at a range of (induction competent) concentrations. However, details on the growth conditions are lacking here. Based on shape of the growth curves, I assume that the conditions were changed (reduced temperature?) upon adding the compounds? And could you test different temperatures, which is a common variable in recombinant protein expression and might affect the toxicity.
- The crystal structure of the repressor EilR with DNA and compounds will allow for future manipulations and optimizations. A description explaining the structural changes upon ligand binding that explain the loss of DNA binding, could aid to this end.

[Reviewers' comments copied verbatim (with issues underlined), interspersed with our responses (*italicized*) and **in bold** where found in the manuscript or supplementary material.]

Reviewer #1 (Remarks to the Author):

The authors develop a bacterial expression system based on the previously identified repressor protein EilR. To this aim they identify a consensus operator sequence and clone this operator in various configurations into existing bacterial promoters. They characterize the performance of this promoter with regard to inducer spectrum (e.g. crystal violet, acridine orange), expression kinetics and dynamic range. They demonstrate that their promoter compares favorably with state of the art expression systems (arabinose system, TET-promoter, LacI). They demonstrate that their promoter performs well in different bacteria such as *P. putida*, *S. meliloti* or *C. crescentus*.

They further solve the structure of EilR and perform mutation studies to identify the molecular basis for DNA-binding and inducer recognition.

The availability of strong bacterial expression systems that are responsive to cheap inducers is of high interest for biotech industry, especially when commodity proteins are to be produced where expensive inducers would negatively impact on the economic feasibility of the process.

In order to be able to better evaluate the potential of this approach but also to get to know potential limitations, it is recommended that the authors address the following points:

1. The favored inducer in the manuscript is crystal violet. Crystal violet is a well-known dye staining bacterial walls (Gram-staining). Furthermore, crystal violet is known to stain DNA and to be mutagenic (see MSDS of crystal violet, PMID 6472325, https://pubchem.ncbi.nlm.nih.gov/compound/Crystal_violet). Such mutagenic activity could induce mutations in the production strain including the target gene sequence and thus lead to product heterogeneity. The authors should experimentally analyze whether crystal violet at concentrations required for maximum gene expression (e.g. 2 μ M, according to Supplementary Fig. 4) induces mutations in the host strain.

We agree that the effects of CV on cells should be considered, so we performed tests on its mutagenicity and stability. Our answer regarding the issue of CV stability in culture is found under the Reviewer's point 4 below.

*For CV mutagenicity, there are a number of reports regarding this and the results vary, depending on the test protocol. Thomas and MacPhee (1984, PMID 6472325, cited by the reviewer) used the Ames method to determine mutagenicity of CV in an *E. coli* strain (in the absence of S9 liver extract), and found the highest number of revertants was at 25 μ g CV per plate: the number of revertants increased three- to five-fold over basal level (for example, from 12 to 55 colonies). The concentration of CV converts to a range of \sim 30 μ M (in top agar) to \sim 2 μ M (whole plate). Ackerman et al. (2009, PMID 19581339) used a bioluminescent *Salmonella* reverse mutation assay to test concentrations to more than 33 μ M and did not find an increase in the number of revertants. The results may be complicated by the sensitivity of test strains to CV.*

If we use the results of Thomas and MacPhee and assume that the concentration of CV tested is similar to the maximum used for our induction experiments (1 μ M), we feel that the level of mutagenesis is negligible for production purposes (at most five-fold over background, increasing from 1×10^{-8} to 5×10^{-8}).

To answer these concerns in our system directly, we designed a new experiment to test mutagenicity. At the optimal level for full expression, 1 μ M CV, we observed a \sim 2-fold increase in rifampicin resistant

colonies in *E. coli* and only a negligible increase in *S. meliloti*. This data is presented in the new **Supplementary Table 1** and stated in the **main text, p. 9**: “Minor mutagenic activities have been reported in different organisms for some of the dyes at higher concentrations; however, the mutagenic effect of the optimal inducer CV at fully inducing levels (0.5 to 1 μM CV) was negligible in the hosts tested, *E. coli* and *S. meliloti*.”

The authors should as well analyze the target gene for mutations after a typical time span required for protein production (e.g. up to 12 h).

CV is known to cause frame-shift type mutations, which we believe are unlikely in our experiments. The reason for this is that we observe high yields of the target protein produced at the expected M_r , as seen in our new results in **Fig. 6a & 6b** and described on **p. 9**. Likewise, both of the expressed proteins are fully functional, based on the fluorescence of RFP and the enzymatic activity of β -galactosidase (see new data in **Materials & Methods, p. 27**). Any frame-shift mutations would result in truncated protein products and potentially loss of activity.

We also feel that any mutations that are favored by basal expression of a toxic protein prior to induction would be minimized by the tightness of our engineered promoters. In new experimental results, we demonstrate the control over both toxic and essential genes, in **Fig. 5a & 5b**, respectively (**pp. 8 & 9**). Induction by 1 μM CV causes high *sacB* expression, which is toxic both in the presence and absence of sucrose (**5a**); repression of the essential gene *pleC* prevents growth in *S. meliloti*, and induction by CV establishes normal growth (**5b**).

2. Similar point: in Supplementary Figure 8 the authors demonstrate that the used inducers negatively impact cell growth (which is also in agreement with previous literature). This seems also the case for crystal violet at concentrations of 2 μM (the concentration where maximum gene expression is obtained, see Supplementary Fig. 4). Is this toxic effect caused by DNA mutations? The authors should accordingly rephrase their statement on Fig. 5, last line.

We showed that CV used at 4 μM has a minor negative effect on *E. coli* (**Supplementary Fig. 6**, formerly 8). Coupled with our mutagenesis testing described in (1), we feel that this statement does not need modification; the higher levels of toxicity by some of the dyes, such as above 4 μM for malachite green and 8 μM for methylene blue are possibly due to mutagenesis, but we did not test for this. With reference to the result in **Supplementary Fig. 3** (formerly 4), doubling the operator sequence causes the higher levels of expression, and 0.5 to 2 μM CV induces the maximum expression observed by fluorescence. We have now indicated this more specifically in the legend.

3. The authors should perform a 1:1 comparison of their system to the most commonly used platforms for the production of recombinant proteins in everyday molecular biology and biotechnology. For example, they should compare their system to IPTG-inducible expression in *E. coli* BL21 (DE3) and *E. coli* BL21 (DE3) pLysS with regard to basal levels, expression kinetics as well as maximally induced levels.

In our original manuscript, former Fig. 3c (now **Fig. 6b**) and Supplementary Fig. 6 (now **Supplementary Fig. 5** that was replotted as a 2D bar chart) illustrate the results of experiments that directly compare the T7 system with Jungle Express in the same standard strain, BL21-DE3. We did not feel that testing the pLysS strain was necessary, because our comparison with the tightly regulated Ptet and PBAD under glucose conditions should be sufficient to demonstrate tightness. In the revised version we have added new results in the **main text, p. 9** and **Fig. 6**. These include side-by-side comparisons of the expression of both RFP and β -galactosidase, including the fractionation of soluble from insoluble proteins and the

measurement of β -galactosidase activity. Based on the staining density of protein bands in the SDS gels, we confirm that our system maintains tight control in un-induced cells, and that after induction the level of both proteins produced is higher than that of the T7 system (**Fig. 6a**). The standard biochemical assay for β -galactosidase indicates the enzymatic activity is about 3-fold higher in protein products of Jungle Express than the T7 system (new data given in **Materials & Methods, p. 27**). [This answers a similar question asked by Reviewer 2 (e)]

In addition, we compared concentrations and costs of inducers of common expression platforms with Jungle Express in **Figure 6c** and **Supplementary Table 2**.

4. What is the stability of crystal violet in culture medium? Is this stability dependent on bacterial density or do bacteria somehow absorb the dye (less likely for Gram-negative bacteria)? If there is an effect of bacterial density: how would this affect performance in routine high density fermentations in which bacterial densities more than 100x higher than reported here are used? Would it then be required to add higher crystal violet concentrations to compensate for the degradation/absorption? Would these concentrations be toxic?

To answer this, we performed an experiment to test CV stability in culture media. We found that CV is stable in growth medium over the 15 h of culture, and that at higher levels the dye is distributed between the medium and cells (new **Supplementary Fig. 7** and main text, **p. 10**). We did not experiment with any high density fermentations.

5. Previously, the crystal structure of a similar repressor protein (RamR) has been reported that binds to similar or the same (e.g. crystal violet) inducers (Yamasaki et al. 2013, doi: 10.1038/ncomms3078). It would be interesting to compare this previous structure to the one of EilR in terms of inducer recognition and DNA-binding properties.

RamR is a TetR-family repressor and binds to the same inducer, so we performed the comparison as suggested. Interestingly, the location of the CV binding site and the surrounding protein structure is significantly different in EilR-CV and RamR-CV, as shown in **Supplementary Fig. 8**. Similar to the EilR-CV binding site, hydrophobic residues surround the RamR-CV binding site; moreover, the overall fold of the RamR DNA-binding domain is similar to EilR, formed by a three-helix bundle containing the HTH motif. However, the residues of EilR that make direct contact with DNA are not conserved in the RamR sequence, indicating different DNA-binding interactions. We include a detailed discussion of this in the legend and in the **main text, p. 10**.

Minor point

6. The 3D representation of Supplementary Figs. 6 and 7 is difficult to interpret (some bars seem hidden behind others). The authors should use a 2D representation where the RFP values for every single condition can clearly be seen. Also, the authors should indicate the number of replicates (for Supplementary Fig. 6) and indicate error bars in both Supplementary Figures.

We have replaced these **Supplementary Figs.**, now **4 & 5**, with a 2D representation as the reviewer suggested, and we indicate any statistical information in the legends.

Reviewer #2

Overview:

In this paper, the authors claim that they have developed an inducible expression system in bacteria with features of tight control and strong expression level, as well as activity in distinct bacterial species, compared to commonly used inducible systems. Also, the authors emphasize the cost

efficiency of the inducer for this expression system, which has the potential for industrial-sized fermentations. A 3-D structure of the repressor protein bound to ligand was solved and presented. Overall, the authors provided data supporting each point they claimed. However, the data were in some cases insufficient and poorly organized and presented. There was a general lack of detailed description and labeling, which caused confusion and raised questions.

What they claimed and what they still need to provide

1. Development of expression system and inducer screen

a) The authors phylogenetically derived a consensus operator sequence for the previously identified repressor EilR, and engineered a series of promoters based on this operator sequence. The biggest problem here is that there is no accession number provided for each operator sequence, nor there an entry for EilR in the protein (NCBI) database nor indeed for the X-ray structure (pdb file). All these sequences must be deposited in the relevant databases to make this paper publishable.

*We have deposited all plasmids in the Joint BioEnergy Institute Public Registry (<https://public-registry.jbei.org>), which will be released upon publication for searching and reviewing the sequences and annotations (**Supplementary Table 4**). Likewise, the EilR sequence and the crystallographic coordinates for EilR and EilR complexes will be released to the NCBI and RCSB PDB, respectively, upon publication.*

b) Another concern here is the engineering process is poorly described. A flow chart is needed to illustrate the construction of the library, the subsequent screen, and the incorporation of Phage promoters. Figure 1c, Method, and Supplementary Figure 2, which are related to this section, are inconsistent.

c) Data from the library screen are missing. RFP expression data for all the colonies picked from the library should be provided.

*These are good points and need clarification. We have now included a complete description of the entire process. In **Fig. 1c**, we indicate a more direct relationship between the consensus operator sequence and its integration into the promoters; supporting this, **Supplementary Fig. 1** gives our initial design of oligonucleotide sequences to amplify the promoter library, the engineering of promoter probe, and our screening protocol and data. We have updated both the **Legends** and **Materials & Methods** to reflect this information.*

2. Regulation evaluation: the authors claimed that EilR system is tight and strong, media-independent, population-uniform and host-versatile.

d) The authors have provided flow cytometry and immunoblot data for RFP expression under regulation of EilR system, indicating a low basal level and high expression with a relatively low concentration of inducer. However, the authors only tested the system with one gene, RFP, which is insufficient. At least one more gene should be tested to be convincing.

e) In addition to flow cytometry, more analysis approaches should be included for the expression level. At minimum some other unrelated protein (lacZ?) should be tested. That's a very easy assay, a substantially larger protein, and should also be evaluated by immunoblot.

We have performed these experiments with β -galactosidase and describe them in our response to Reviewer 1 (3) above.

f) The authors claimed the EilR system is media-independent, but failed to provide the analysis data for different culture media.

We now present results in **Fig. 3** and on **p. 7** to compare RFP expression as a function of the type of promoter in *E. coli* grown in both defined media (EZ-Rich with glucose) and in complex media (LB). Our results show that the growth media do not affect the activity of PJEx and PTet promoters, whereas the choice of media substantially affects both basal and induced expression from PBad and PTrc.

3. Comparison to common inducible promoters

Although the reviewer's question in this case is not clear, please see comments in (f) above. Also, we demonstrate in **Fig. 3** the comparisons of PJEx in *E. coli* to PTet, PBad, and PTrc, and the comparison with PT7/LacI in **Fig. 6** and in **Supplementary Fig. 5**. In *P. putida*, *S. meliloti*, and *C. crescentus*, comparisons are with the TetR-regulated PLtetO-1, shown in **Fig. 4**.

Reviewer #3

In this study, the authors describe the development of a new method to induce protein over-expression in bacteria. It is based on their previously discovered bacterial multidrug efflux system. Normally, this system controls the tolerance to environmental (toxic) compounds, where the compounds directly de-repress of the expression of membrane transporters. The authors identified the cognate repressor (EilR) DNA binding sequence, and use this to design and create a palindromic consensus operator that shows a strong and improved binding affinity to the repressor. By combining this DNA sequence with commonly used viral promoters (that are recognized by the host polymerases), they created an expression system that allows high level protein production with precise time control, which they named Jungle Express (JEx).

The system does rely on the addition of natural toxic compounds to the medium. However, the sensitivity of this system allows the use of very low concentrations of cheap inducers that are tolerated by the cells. This is a key feature, because it would allow the induction of large scale cultures, something that is problematic with existing systems. Additionally, JEx does differentially respond to a range of compounds, allowing for tuning the induction for your needs.

Together this system promises high-level and tunable protein expression in bacteria, and adds a couple of features that set it apart from widely used analogous system. Overall, I believe the study is well executed and will find an audience in both research and industry.

Minor points:

- What are the respective concentration of the binding partners in gel shift assays (see for example Fig 1b, S1 and S9)?

We now provide the concentrations of binding partners and ratios of repressor protein to operator DNA in the legends to these figures. To illustrate the EMSA results originally shown in Supplementary Fig. 1, **Fig. 1b** has been modified to contain this and other information, while at the same time removing redundancies.

- What is the unit of the concentration of [C2C1mim]Cl in figure S2d?

We have corrected this - the concentration on the x-axis of **Supplementary Fig. 1f** is in mM.

- A potential drawback of the system is the use of toxic compounds. The authors do address this point in figure S8, where they concluded that a few compounds did not affect cell growth at a range of (induction competent) concentrations. However, details on the growth conditions are lacking

here. Based on shape of the growth curves, I assume that the conditions were changed (reduced temperature?) upon adding the compounds? And could you test different temperatures, which is a common variable in recombinant protein expression and might affect the toxicity.

Temperatures for the toxicity assay in **Supplementary Fig. 6** (formerly 8) were standard for *E. coli* (37 °C) upon adding inducers; we have now indicated this in the figure legend. Kinetic plate reader measurements were started after induction at early exponential phase, hence the lack of a lag phase in the growth curves. As mentioned in the legend, values were normalized to the OD₆₀₀ at the start of the measurement to account for intrinsic dye absorbances at 600 nM. We tested 18 and 30 °C in addition to 37 °C in the measurements represented in **Supplementary Fig. 5** (previously 6), comparing RFP expression from PJEx with PT7.

The crystal structure of the repressor EilR with DNA and compounds will allow for future manipulations and optimizations. A description explaining the structural changes upon ligand binding that explain the loss of DNA binding, could aid to this end.

We now provide additional structural details to clarify the ligand binding in EilR in the **main text, p. 11**, and in the Supplementary Section. **Supplementary Fig. 10** indicates the contacts of CV and MG in the ligand binding pocket, and the effect of substituting negatively charged residues, which is complementary with EMSA results from the operator DNA binding with EilR and the mutations in EilR that affect binding (**Supplementary Fig. 9**). In addition, in **Supplementary Fig. 11** we compared the influence of CV and MG binding on the conformational changes in DNA binding domains. The discussion of the repressor-operator interaction is now expanded in the **main text, p. 12**.

REVIEWERS' COMMENTS:

Reviewer #1 (Remarks to the Author):

For this revised version the authors have performed additional experiments and amended the manuscript to address the reviewers' comments. Especially they now provide data on the possible mutagenic effect of the inducer and demonstrate that mutagenicity can likely be neglected at the inducer concentrations used.

The amended manuscript adequately addresses my concerns. This expression system will very likely be of high value for the recombinant production of proteins in bacteria in both, lab scale as well as industrial large scale.

Reviewer #3 (Remarks to the Author):

I thank the authors for their revisions and am satisfied that they have answered my criticisms.